# SOCS7-Derived BC-Box Motif Peptide Mediated Cholinergic Differentiation of Human Adipose-Derived Mesenchymal Stem Cells

**DOI:** 10.3390/ijms24032786

**Published:** 2023-02-01

**Authors:** Hiroshi Kanno, Shutaro Matsumoto, Tetsuya Yoshizumi, Kimihiro Nakahara, Masamichi Shinonaga, Atsuhiko Kubo, Satoshi Fujii, Yasuyuki Ishizuka, Masaki Tanaka, Masamitsu Ichihashi, Hidetoshi Murata

**Affiliations:** 1Department of Neurosurgery, Yokohama City University, Yokohama 236-0004, Japan; 2Department of Neurosurgery, Asahi Hospital, Tokyo 121-0078, Japan; 3Department of Neurosurgery, International University of Health and Welfare Atami Hospital, Atami 413-0012, Japan; 4Department of Neurosurgery, St. Marianna Medical University of Medicine, Kawasaki 216-8511, Japan; 5Nerve Care Clinic, Yokosuka 238-0012, Japan; 6BioFuture Technologies Ltd., Tokyo 105-0014, Japan; 7BTR Arts Ginza Clinic, Tokyo 105-0004, Japan

**Keywords:** adipose-derived mesenchymal stem cells, SOCS7-derived BC-box motif peptide, cholinergic differentiation, ubiquitination

## Abstract

Adipose-derived mesenchymal stem cells (ADMSCs) are a type of pluripotent somatic stem cells that differentiate into various cell types such as osteoblast, chondrocyte, and neuronal cells. ADMSCs as donor cells are used to produce regenerative medicines at hospitals and clinics. However, it has not been reported that ADMSCs were differentiated to a specific type of neuron with a peptide. Here, we report that ADMSCs differentiate to the cholinergic phenotype of neurons by the SOCS7-derived BC-box motif peptide. At operations for patients with neurological disorders, a small amount of subcutaneous fat was obtained. Two weeks later, adipose-derived mesenchymal stem cells (ADMSCs) were isolated and cultured for a further 1 to 2 weeks. Flow cytometry analysis for characterization of ADMSCs was performed with CD73, CD90, and CD105 as positive markers, and CD14, CD31, and CD56 as negative markers. The results showed that cultured cells were compatible with ADMSCs. Immunocytochemical studies showed naïve ADMSCs immunopositive for p75NTR, RET, nestin, keratin, neurofilament-M, and smooth muscle actin. ADMSCs were suggested to be pluripotent stem cells. A peptide corresponding to the amino-acid sequence of BC-box motif derived from SOCS7 protein was added to the medium at a concentration of 2 μM. Three days later, immunocytochemistry analysis, Western blot analysis, ubiquitination assay, and electrophysiological analysis with patch cramp were performed. Immunostaining revealed the expression of neurofilament H (NFH), choline acetyltransferase (ChAT), and tyrosine hydroxylase (TH). In addition, Western blot analysis showed an increase in the expression of NFH, ChAT, and TH, and the expression of ChAT was more distinct than TH. Immunoprecipitation with JAK2 showed an increase in the expression of ubiquitin. Electrophysiological analysis showed a large holding potential at the recorded cells through path electrodes. The BC-box motif peptide derived from SOCS7 promoted the cholinergic differentiation of ADMSCs. This novel method will contribute to research as well as regenerative medicine for cholinergic neuron diseases.

## 1. Introduction

Adipose-derived mesenchymal stem cells (ADMSCs) are a type of pluripotent somatic stem cells that differentiate into various cell types such as osteoblast, chondrocyte, and neuronal cells. In a previous study [1], it was reported that ADMSCs express various proteins, including nestin, YAP, AMOT, β-tubulin III, GFAP, and NeuN, and genes, including *OCT4*, *NANOG*, *FGF2*, *REX1*, *VEGF*, *ABCG2*, *TUBB3*, *MAP2*, *NEFM*, *SOX2*, *NES*, *SLC1A3*, *S100*, and *ASCL1*. So far, many clinical trials using ADMSCs in regenerative medicine have been performed for not only knee arthritis but also neuronal diseases such as stroke, spinal cord injury, and neurodegenerative diseases [2,3,4,5,6]. In addition, they have been clinically applied in human patients [7]. The ADMSCs are isolated from subcutaneous adipose tissue that differentiate into various cell types such as chondrocytes, osteoblasts, and neurons [8]. It has been demonstrated that ADMSCs differentiate into chondrocytes in knee joints [9]; however, it is not fully elucidated that ADMSCs are able to differentiate into neurons in the central nervous system. Nevertheless, it has been reported that ADMSCs improve recovery after stroke [10]. It is unknown how ADMSCs contribute to neuronal regeneration in the central nervous system of patients with neuronal diseases. The speculated mechanisms of how ADMSCs contribute to neuronal regeneration have been proposed as follows: (1) trans-differentiation to neuronal cells at lesion site; (2) immunomodulation by AMDSCs; and (3) stimulation of neuronal regeneration by trophic factors secreted from ADMSCs [10]. If trans-differentiation from ADMSCs to neuronal cells occurs, neuronal differentiation of ADMSCs before transplantation to patients with neuronal diseases will be desirable. Furthermore, trans-differentiation of ADMSCs to specific phenotypes of neurons will be more desirable for specific neuron diseases. Following this point of view, we propose a novel method of trans-differentiation of ADMSCs to a specific phenotype of neurons. 

The suppressor of cytokine signaling (SOCS) proteins act as negative feedback inhibitors. The SOCS box-containing proteins are subclassified into CIS and SOCS1–SOCS7 proteins based on the domains contained in their central regions [11]. The SOCS box motif recruits an E3 ubiquitin ligase complex consisting of Elongin B and C, Rbx2, and Cullin5. This complex forms the E3 ubiquitin ligase which tags target proteins such as the Janus kinases (JAKs) and cytokine receptors with ubiquitin, marking them for proteasomal degradation [12,13]. Ubiquitination of a substrate via SOCS proteins can lead to its proteasomal degradation. So far, numerous different molecules such as interferons, a large number of cytokines, growth factors, and hormonal factors are known to activate the JAK-STAT (signal transducer and activator of transcription) pathway, which is inhibited by SOCS proteins. 

SOCS proteins are associated with the differentiation of various cell types. They also play important roles in neuronal differentiation, neurogenesis, nervous system development, and nerve regeneration. Among the members of this family, SOCS1, SOCS2, and SOCS3 are expressed in the nervous system throughout development [14]. SOCS1 regulates the interferon gamma-mediated survival of sensory neurons [15] and regulates the proliferation and neuronal differentiation of neural stem cells (NSCs) [16,17]. SOCS2 was also demonstrated to promote neurite outgrowth, regulate neuronal morphology, induce neurogenesis, and inhibit astrogliogenesis of NSCs [18,19]. SOCS3 induces neuronal differentiation and promotes neural cell survival in PC12 cells [20]. In addition, it was found that SOCS6 also promotes neuronal differentiation of NSCs [21]. Recently, we demonstrated that the BC-box motif in SOCS6 induces GABAnergic differentiation of epidermal stem cells [22]. SOCS proteins are associated with neuronal diseases such as Alzheimer’s disease (AD), Parkinson’s disease (PD), multiple sclerosis (MS), amyotrophic lateral sclerosis (ALS), ischemic stroke (IS), and traumatic brain injury (TBI) [23,24,25,26,27,28,29,30,31,32]. There are significantly increased levels of SOCS4 and SOCS7 proteins in AD brains [33]. It has been proposed that SOCS6 and SOCS7 bind tyrosine-phosphorylated DAB1 and commit it to ubiquitination and proteasomal degradation. Removal of phosphorylated DAB1 by SOCS6 and SOCS7 terminates Reelin signaling, allowing commencement of a new cycle of Reelin signaling. In addition, SOCS7 controls Muller glia cell lamination; it is not responsible for cone photoreceptor positioning, suggesting that RBX2, most likely through CRL5 activity, controls other signaling pathways required for proper cone localization [33,34,35]. So far, we have demonstrated that BC box motif peptides in some BC-box proteins, such as the von Hippel–Lindau (VHL) protein, promote the neuronal differentiation of somatic stem cells [36]. However, the mechanism of neuronal differentiation by BC-box motif in SOCS7 has not yet been fully elucidated. Here, we show that the BC-box motif peptide derived from SOCS7 induces cholinergic neuronal differentiation of ADMSCs via inhibition of the JAK2/STAT3 pathway. Furthermore, we propose that cholinergic neuron-like cells would be promising for neuronal regeneration for motor neuron disease. In addition, we discuss the mechanism of neuronal differentiation via the BC-box motif peptide derived from SOCS7.

## 2. Results

### 2.1. Isolation and Proliferation of Adipose-Derived Mesenchymal Stem Cells

At surgeries, 0.5 to 1.0 cm^3^ adipose tissue was obtained from abdominal or gluteal subcutaneous tissue of eleven patients with neuronal diseases (pyriformis muscle syndrome; hydrocephalus). Out of eleven, seven patients with pyriformis muscle (mean age, 43.5; all female) underwent surgery for dissection of sciatic nerve by the resection of the pyriformis muscle in the gluteal region, and the remaining four patients with hydrocephalus (mean age, 72.3; 2 females and 2 males) underwent ventricle-peritoneal shunt surgery or lumbo-peritoneal shunt surgery. On a clean bench in the experimental room, the tissues were finely cut in 1 mm^3^ size and put on the adipose-derived stem cell separating hydroxyapatite-coated polyethylene-polypropylene (PEPP) seat (BioFuture Technologies Ltd., Tokyo, Japan) [37] in a 6 cm diameter culture dish containing 6 mL of adipose-derived mesenchymal stem cells (ADMSCs) isolation medium (BioFuture Technologies Ltd.) with 1% FCS. After 10 to 14 days, ADMSCs were isolated and attached to filaments in a hydroxyapatite-coated PEPP seat. After the cells were treated with 0.25% trypsin-EDTA solution and separated into single cells, they were adhesively cultured in a 90 mm dish. At this stage, the cells showed a round or elliptical shape. Three days after the start of adhesive culture, the cells changed to a triangular or square shape with or without a long process like a neurite. Seven days later, the cells were cultured semi-confluently, and two weeks later they were layered (Figure 1).

### 2.2. Characterization of ADMSCs with Flowcytometry and Immunocytochemistry

To characterize naïve ADMSCs, a flow cytometry (fluorescence-activated cell sorting, FACS) analysis and an immunocytochemical study were performed. FACS data of ADMSCs showed CD73-positive, 96.6%; CD90-positive, 93.16%; CD105-positive, 87.82%; CD31-positive, 1.18%; CD34-positive, 3.52%; and CD56-positive, 1.52%. The data revealed that ADMSC-positive markers (CD31, CD90, and CD105) were reactive in very high level, while ADMSC-negative markers (CD31, Cd34, and CD56) were reactive in very low level. Thus, FACS-analyzed cells were compatible with ADMSCs (Figure 2) [38,39].

The immunocytochemical study showed that naïve ADMSCs were immunopositive for p75 NTR (neurotrophin receptor, a mesenchymal precursor marker); for ret proto-oncogene product (RET, an endodermal precursor marker); for nestin (a neural progenitor marker); and for keratin (an epidermal stem cell marker). We also found small fractions of the cells immunoreactive with antibodies against neurofilament triplet M (NF-M, a neuron marker), and smooth muscle actin (SMA, a smooth muscle cell marker). These results suggested ADMSCs to be pluripotent stem cells (Figure 3). 

### 2.3. Neuronal Differentiation with SOCS7-Derived BC-Box Motif Peptide

SOCS7-derived BC-box motif peptide conjugated with protein transduction domain (SOCS7 peptide) was chemically synthesized. The amino-acid sequence of the SOCS7-derived peptide was YARAAARQARASLQYLCRFVIRQYTR (single underline, PTD; double underline, BC box motif). The SOCS7 peptide was added into ADMSC-proliferation medium at 2 mM of concentration as described previously [22,40]. Then, neuronal differentiation started 4 h after addition. 

### 2.4. Immunocytochemistry for ADMSCs with or without the SOCS7 Peptide Treatment

To evaluate the immunoreactivity of ADMSCs with or without the SOCS7 peptide treatment, immunostaining for ADMSCs was performed by using anti-neurofilament-H (NFH), anti-acetyl choline transferase (ChAT), and anti-tyrosine hydroxylase (TH) antibodies as the primary antibodies. We employed biotinylated anti-IgG antibody as the secondary antibody. The immunocytochemical study on the SOCS7-peptide-treated cells showed the distinct expressions of mature neuronal marker NFH, cholinergic marker ChAT, and dopaminergic marker TH, while non-treated cells showed negative expression of these markers. These results indicated cholinergic or dopaminergic differentiation (Figure 4). 

### 2.5. Western Blotting 

To evaluate protein expressions of ADMSCs with or without SOCS7-peptide treatment, western blot analysis was performed. Western blots were probed with anti-NFH, anti-nestin, anti-ChAT, anti-MAP2, anti-TH, and βⅢ-Tublin followed by horseradish peroxidase-conjugated secondary antibodies. Western blotting revealed that the levels of ChAT, MAP2, βⅢ-Tublin, and NFH were markedly increased, and TH was slightly increased in the SOCS7-treated cells, while nestin was decreased in the SOCS7-treated cells. This result indicated that SOCS7-peptide-treated cells showed neuronal differentiation to the cholinergic phenotype rather than dopaminergic phenotype (*p* < 0.05) (Figure 5). 

By ubiquitination assay, we investigated whether JAK2 was ubiquitinated after intracellular delivery of the SOCS7 peptide. Immunoprecipitation with anti-JAK2 antibody followed by immunoblotting with anti-ubiquitin antibody revealed more distinct expression in SOCS7-treated ADMSCs than non-treated cells. This result indicated that JAK2 was bound to ubiquitin following treatment of SOCS7 peptide in ADMSCs. These experiments were done at least twice. These results suggested that the SOCS7 peptide-mediated neuronal differentiation of ADMSCs was related to JAK2 ubiquitination followed by inhibition of the JAK2-STAT3 pathway [40] (Figure 5). 

### 2.6. Electrophysiological Analysis by Use of the Patch-Cramp Configuration 

We electrophysiologically analyzed the SOCS7-peptide-mediated neuronal differentiation of ADMSCs. Then, we recorded voltage-gated inward and outward currents in the whole-cell patch-clamp configuration. In whole-cell recordings of 2 µM SOCS7-peptide-delivered ADMSCs, the depolarizing voltage steps elicited both large outward potassium currents and fast inward Na^+^ currents, which are hallmarks of differentiated neurons. Conversely, in the whole-cell recording of non-treated naïve ADMSCs, neither outward potassium nor inward Na^+^ currents were elicited [41] (Figure 6).

## 3. Discussion

Adipose tissue-derived mesenchymal stem cells (ADMSCs) are the ideal somatic stem cell population that can be derived in an autologous fashion from small amounts of accessible subcutaneous tissue biopsies. The present study suggested ADMSCs are pluripotent stem cells. They are capable of differentiating into both neural and mesodermal cells. Regenerative medicines employing ADMSCs are used to treat knee osteoarthritis [2] or neuronal diseases such as stroke [3] and spinal cord injury [4]. Histological analyses have demonstrated that intra-articular administration of human ADMSCs contributes to hyaline-like cartilage regeneration [10]. On the other hand, histological analysis of human ADMSCs that were intravenously administrated to stroke- or spinal-injury patients has not been done. ADMSCs for neuro-regeneration have been administrated intravenously, intrathecally, or into brain or spinal cord parenchyma. Human ADMSCs have been mostly administered intravenously and partially intrathecally, whereas they have been scarcely done into brain or spinal cord parenchyma. Intravenous administration of ADMSCs contributes to neuronal regeneration for neuronal diseases, which has been demonstrated in animal models [42]. However, it has not been demonstrated that intravenously administrated ADMSCs migrate to the animal brain. It seemed that ADMSCs secreted various factors that play roles in neuronal regeneration for neuronal disease animal models [43,44]. On the other hand, before cell transplantation into brain or spinal cord parenchyma or cerebrospinal fluid cavity for regeneration-aimed cell-based therapy for neuronal diseases, neuronal differentiation of the donor cells is fundamental because untreated naive cells scarcely differentiate into neurons in engrafted neural tissues. However, efficient regeneration of neurons from those stem cells and also differentiation to cholinergic neuron phenotypes from those cells has scarcely been demonstrated [45,46]. Therefore, an efficient neuronal differentiation method is required for the study of the differentiation of ADMSCs into the cholinergic neuron phenotype. Moreover, an induction method of neuronal differentiation to the cholinergic neuron phenotype proposed by us can also be useful in treating cholinergic neuron diseases. For example, the cholinergic neuron phenotype will be useful for in vitro drug assays for cholinergic neuron diseases.

Previously, we demonstrated the generation of dopaminergic neuron phenotype or GABAnergic neuron phenotype from neural stem cells or epidermal stem cells via the intracellular delivery of a BC-box motif peptide derived from SOCS6 or VHL [22,36]. In the present study, we proposed a novel method using the intracellular delivery of a BC-box motif peptide derived from SOCS7 for the cholinergic differentiation of ADMSCs. The mechanism of SOCS7 peptide-mediated neuronal differentiation was also suggested to be associated with the inhibition of the JAK/STAT pathway after the SOCS7 peptide bonded to elongin C (Figure 7).

Our employed method of direct intracellular transfer of a synthesized peptide conjugated with a protein transduction domain was previously reported in the literature [47]. It brings about rapid transfer as well as rapid differentiation. For intracellular protein expression, it seems that the intracellular delivery of a peptide conjugated to a protein transduction domain is superior to gene transfer, which is accompanied by problems such as troublesome handling, toxicity, carcinogenicity, and infectious risk with viral vectors. In this study, using morphological, immunocytochemical, electrophysiological, and western blot methods, we observed the neuronal differentiation of ADMSCs into the cholinergic phenotype with the BC-box motif peptide derived from SOCS7 protein. Through the immunohistochemistry and western blotting, recognized TH expression may be suggested to be a near relativity between cholinergic neurons and dopaminergic neurons. Previous reports demonstrated that mesenchymal stem cells were successfully differentiated into cholinergic neurons using tricyclodecane-9-yl-xanthogenate (D609), a specific inhibitor of phosphatidylcholine-specific phospholipase C [46,48]. D609 treatment on MSCs is a simple and rapid method for cholinergic neuron induction as our present method. 

Our proposed novel method of neuronal differentiation of ADMSCs into the cholinergic phenotype using the SOCS7 peptide will contribute to research as well as regenerative medicine for the treatment of cholinergic neuron disease. 

## 4. Experimental Section

### 4.1. Isolation of Adipose-Tissue Derived Mesenchymal Stem Cells from Fat Tissue and Cell Culture

Small volumes of subcutaneous adipose tissues were obtained at surgeries. On a clean bench in the experimental room, the adipose tissues were finely cut in 1 mm^3^ size and put on the adipose-derived stem cell separating hydroxyapatite-coated polyethylene-polypropylene (PEPP) seat (BioFuture Technologies Ltd., Tokyo, Japan) [38] in a 6 cm diameter culture dish containing 6 mL of adipose-derived mesenchymal stem cells (ADMSCs) isolation medium (BioFuture Technologies Ltd.) with 1% FCS (Figure 1A). After 10 to 14 days, cultured adipose-derived mesenchymal stem cells were washed with 0.25% trypsin-EDTA solution with phenol red, centrifuged at 1500 rpm/min for 5 minutes, filtrated with a cell strainer, and cultured in a 9 cm diameter dish containing 10 mL of ADMSCs proliferation medium (BioFuture Technologies Ltd.) for one to two weeks. The number of ADMSCs amounted from 50,000,000 to 150,000,000 and a part of them was used for the following experiments.

### 4.2. Flow Cytometry Analysis for Characterization of Isolated Adipose-Derived Mesenchymal Stem Cells

For flow cytometry (fluorescence-activated cell sorting, FACS) analysis, 1 × 10^6^ ADMSCs were used to detect the expression of cell surface markers: CD73, CD90, and CD105 as positive markers, and CD14, CD31, and CD56 as negative markers. After trypsinization, ADMSCs were counted, centrifuged, and washed twice with phosphate-buffered saline (PBS). They were then stained with the following antibodies: PE (phycoerythrin)-conjugated against CD73, CD90, CD105, CD14, CD31, CD56, and IgG2a antibodies (Miltenyi Biotec, Bergisch Gladbach, Germany). After antibodies were appropriately diluted, cells were vortexed, incubated at room temperature in the dark for 10 min, washed twice with PBS, and centrifuged. The supernatant was decanted, and the cells were resuspended in 300 µL of PBS for FACS analysis. Experimental settings were set up using unstained cells, single color stain, and fluorescence minus one control. A minimum of 10,000 events was recorded. FACS analysis was performed with Guava^®^ easyCyte™ 8 (Luminex, Austin, TX, USA), and Guava software (Luminex) was used for FACS data analysis.

### 4.3. BC-Box Motif Peptide Design and Synthesis

BEX. Co., Ltd. (Tokyo, Japan) was commissioned to do the peptide synthesis. The SOCS7-derived peptide comprises the BC box motif ((A, P, S, T) LXXX (A, C) XXX (A, I, L, V)) corresponding to the binding site of elongin C and five amino acids at the C-terminus of SOCS7 (SLQHLCRFRI-RQLVR). To facilitate the intracellular entry of the synthesized peptide, we employed the protein transduction domain (PTD)-mediated peptide delivery system, by which these peptides were conjugated with PTD consisting of a modified TAT (YARAAARQARA). The amino-acid sequence of the SOCS7-derived peptide was YARAAARQARASLQYLCRFVIRQYTR (single underline, PTD; double underline, BC box motif) (Figure 7A) [22].

### 4.4. Neuronal Induction with BC-Box Motif in SOCS7 Peptide

Semi-confluent or confluent cultured ADMSCs were treated with 0.25% trypsin-EDTA solution and separated into single cells and were put on cover glasses in a 6-well dish. Then, the BC-box motif in the SOCS7 peptide was added into culture medium at 2 mM concentration. The peptide penetrates into cells in 60 min, and neuronal differentiation commences [22].

### 4.5. Immunocytochemistry 

Immunostaining for naïve ADMSCs was performed by using anti- p75NTR (1:100; Sigma-Aldrich, St. Louis, MA, USA), anti-RET (1:100; LBSBio, Seattle, WA, USA), anti-nestin (1:200, Irvine, CA, USA), anti-keratin 15(1:200; LBSBio), anti-neurofilament-M (1:100; Chemicon, Temecula, CA, USA), and anti-smooth muscle actin (SMA; 1:200; Sigma-Aldrich) antibodies as the primary antibodies. The following secondary antibodies were used: rhodamine-conjugated anti-IgG (1:100; Sigma-Aldrich) or FITC-conjugated anti-IgG (1:100; Sigma-Aldrich). Observations were made with a fluorescence microscope system (FV300, Olympus, Tokyo, Japan).

In addition, immunostaining for ADMSCs with or without the SOCS7 peptide treatment was performed by using anti-neurofilament-H (NFH) (1:100; Sigma-Aldrich, St. Louis, MA, USA), anti-ChAT (1:100; Chemicon, Temecula, CA, USA), and anti-TH (1:200; Sigma-Aldrich, St. Louis, MA, USA) antibodies as the primary antibodies. We employed biotinylated anti-IgG antibody as the secondary antibody. Following incubation for 30 min, avidin-biotinylated peroxidase complex was added and incubated for 30 min. Then, diaminobenzidine was used as the substrate, and hematoxylin was used for the counterstaining. 

### 4.6. Western Blotting

Cultured cells were washed three times with cold PBS and then scraped into ice-cold PBS. After incubation on ice for 10 min, the cells were lysed with lysis buffer and centrifuged, after which the supernatants were collected. Each sample was separated by SDS-PAGE under reducing conditions and transferred electrophoretically to nitrocellulose filters. Non-specific binding of antibodies was blocked by incubation with 5% donkey serum for 1 h. Western blots were probed with anti-neurofilament-H (NFH) (1:500; Sigma-Aldrich, St. Louis, MA, USA), anti-ChAT (1:500; Chemicon, Temecula, CA, USA), anti-TH (1:500; Sigma-Aldrich, St. Louis, MA, USA), anti-MAP2 (1:500; GeneTex, Irvine, CA, USA), anti-nestin (1:500; GeneTex), anti-βⅢ-Tublin (1:500; GeneTex), and anti-JAK2 (1:500; Santa Cruz Biotechnology, SCB, San Diego, CA, USA) followed by horseradish peroxidase-conjugated secondary antibodies. Following incubation for 30 min, avidin-biotinylated peroxidase complex was added and incubated for 30 min. Then, protein bands were detected by using 3,3’,5,5’-tetramethylbenzidine (WSE-7140 EzWestBlue W, ATTO CORPORATION) as the substrate. 

### 4.7. Ubiquitination Assay 

Total protein was extracted from the cells, and the lysates were immunoprecipitated with anti-JAK2 antibody (SCB)using Protein A/G (BioVision, Milpitas, CA, USA). Further, each sample was separated by SDS-PAGE and transferred electrophoretically to nitrocellulose filters. Western blots were probed with anti-ubiquitin antibody (Sigma-Aldrich). Additionally, to examine the possible inhibition of the JAK2/STAT3 pathway, we probed the western blots with anti-JAK2 (1:500; SCB) [39].

### 4.8. Electrophysiology with Patch-Cramp Configuration 

To record fast sodium and delayed rectifier potassium currents, we prepared extracellular and intracellular solutions as described previously [21,22,39,40]. Five days after the addition of SOCS7 peptide at a 2 µM concentration, a holding potential of −80 mV and voltage step of 20 mV over the range of −100 to 100 mV with 50-ms durations were applied to the recorded cells through patch electrodes. For recordings and data analysis, we used CEZ-2300 (Nihon Kohden, Tokyo, Japan) and pCLAMP 6.0 software (Axon Instruments, Burlingame, CA, USA). Linear components of leak and capacitive currents were reduced by analogue circuitry and then canceled by the P/N method. Signals were sampled every 20 µsec, and currents were filtered at 5 kHz. Data were additionally processed with Origin 5.0 (Microcal, Northhampton, MA, USA) [40].

### 4.9. Statistics

Numerical data were presented as the mean (%) ± SEM. Factorial analysis of variance applied to each group with pairwise comparison done by the Bonferroni method or Mann–Whitney U-test was noted. To verify whether differences between distinct conditions reached the significance level, we used *p* < 0.05. 

## Figures and Tables

**Figure 1 ijms-24-02786-f001:**
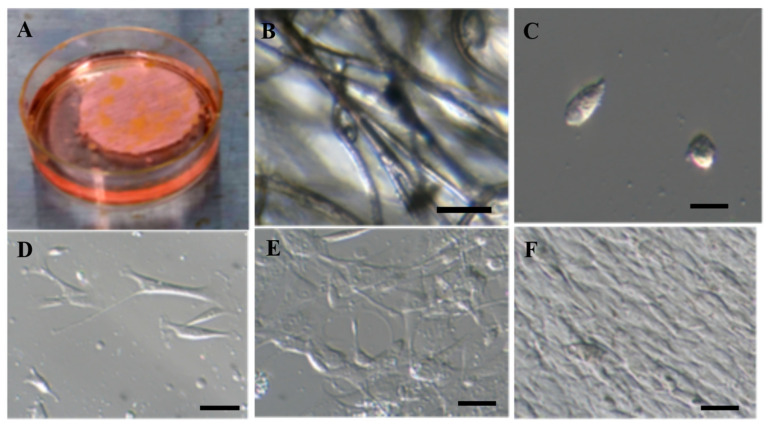
Isolation and culture of adipose-derived mesenchymal stem cells (ADMSCs). (**A**) Adipose-tissue fragments on a polyethylene polypropylene (PEPP) seat. (**B**) ADMSCs attach to fibers in the PEPP seat. Red arrows, ADMSCs. (**C**) Adipose-derived mesenchymal stem cells (ADMSCs) immediately after being isolated from a PEPP seat. (**D**) Three days after the isolation of ADMSCs. (**E**) Semi-confluent condition at 10 days after the isolation of ADMSCs. (**F**) Confluent condition at 14 days after the isolation of ADMSCs. Scale bar, 10 μm.

**Figure 2 ijms-24-02786-f002:**
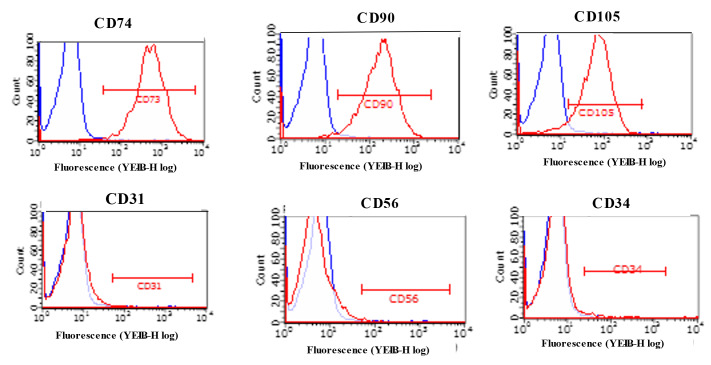
Flow cytometry analysis for isolated adipose-derived stem cells. Red curves PE-labeled against surface markers (CD74, CD90, CD105, CD31, CD56, CD34) antibodies. Blue curves, PE-labeled against IgG2a antibodies. Upper row, PE-labelled against CD74, CD90, and CD105 antibodies showing completely different curves from PE-labelled against IgG2a. Lower row, PE-labelled CD31, CD56, and CD34 antibodies showing curves nearly overlapping PE-labelled against IgG2a.

**Figure 3 ijms-24-02786-f003:**
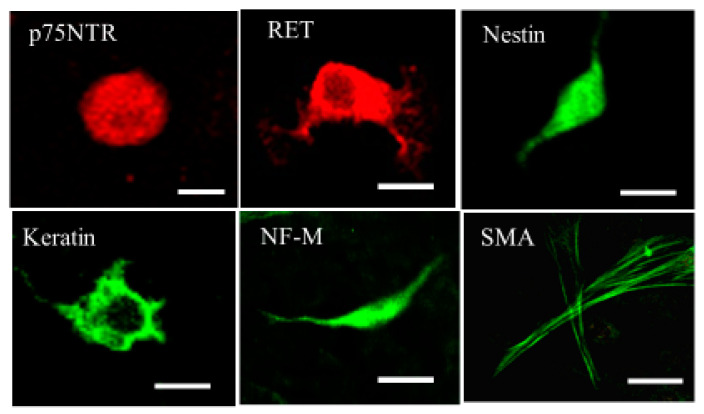
Immunofluorescent images of naïve ADMSCs. p75NTR, p75 neurotrophin receptor, a mesenchymal precursor marker; RET, ret proto-oncogene product, an endodermal precursor marker; nestin, a neural progenitor marker; keratin, an epidermal stem cell marker; NF-M, neurofilament-M, a neuron marker; SMA, smooth muscle actin, a smooth muscle cell marker. Fluorescein isothiocyanate (FITC), green; rhodamine, red. Scale bar = 10 μm.

**Figure 4 ijms-24-02786-f004:**
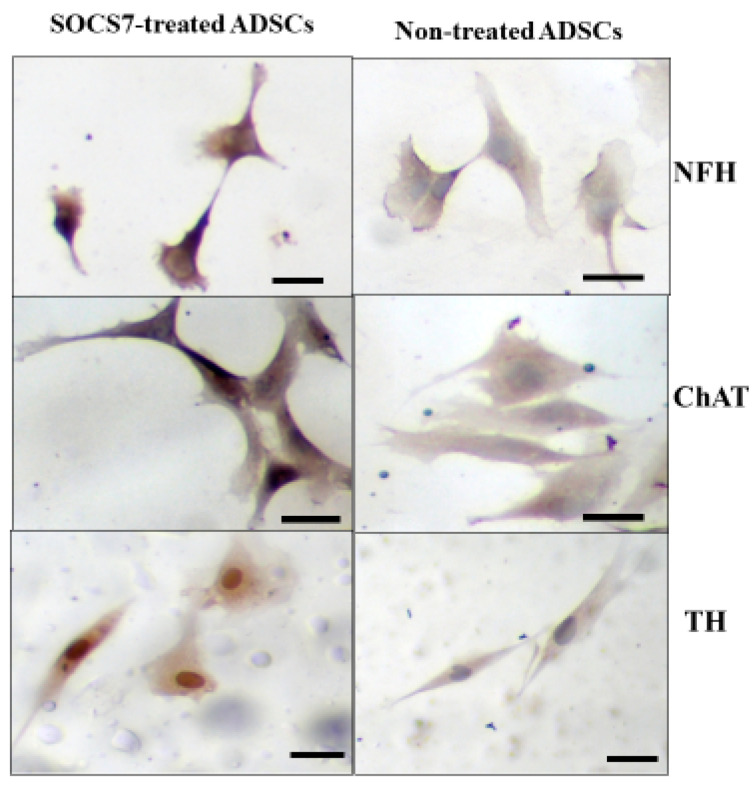
Immunocytostaining with anti-Neurofilament-H (NFH), choline acetyl transferase (ChAT), tyrosine hydroxylase (TH) antibodies for SOCS7-peptide-treated ADMSCs and non-treated ADMSCs. SOCS7-peptide treated cells showed immunoreactive to NFH, ChAT, and TH, while non-treated cells showed negative. Scale bar = 10 μm.

**Figure 5 ijms-24-02786-f005:**
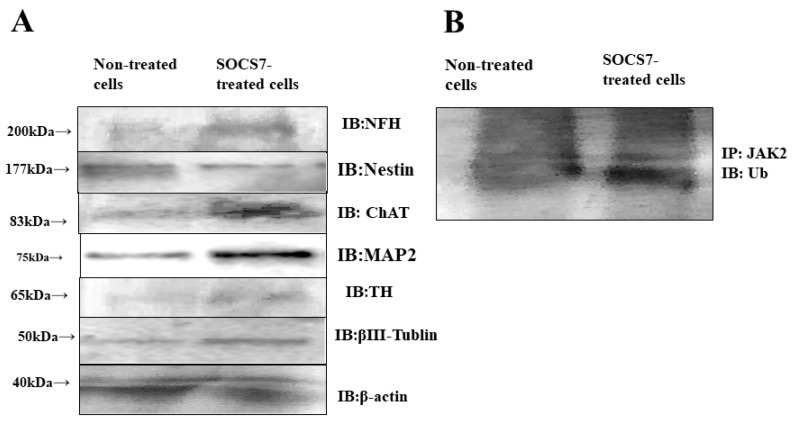
Immunoblotting (IB) and immunoprecipitation (IP) study on SOCS7-peptide-treated and non-treated cells. (**A**) Immunoblotting was done with anti-neurofilament-H (NFH), choline acetyl transferase (ChAT), nestin, choline acetylcholine transferase (ChAT), microtubule-associated protein 2 (MAP2), tyrosine hydroxylase (TH), and βⅢ-Tublin antibodies, anti-NFH antibody, anti-ChAT antibody, anti-TH antibody, and anti-actin antibody as an internal control. (**B**) Immunoprecipitation with anti-JAK2 antibody was followed by immunoblotting with anti-ubiquitin antibody.

**Figure 6 ijms-24-02786-f006:**
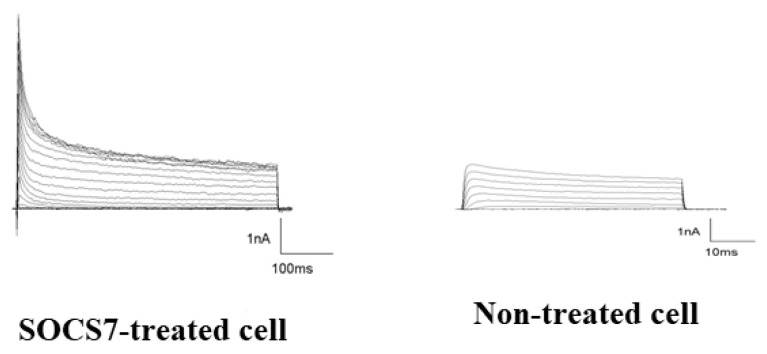
Electrophysiological properties of peptide-treated cells. Voltage-gated inward and outward currents were recorded in the whole-cell path-clamp configuration. Left, a SOCS7-peptide-treated cell elicited both large outward K^+^ currents and fast inward N^+^ Current. Right, a non-treated cell elicited small outward K^+^ currents and minute fast Na+ currents.

**Figure 7 ijms-24-02786-f007:**
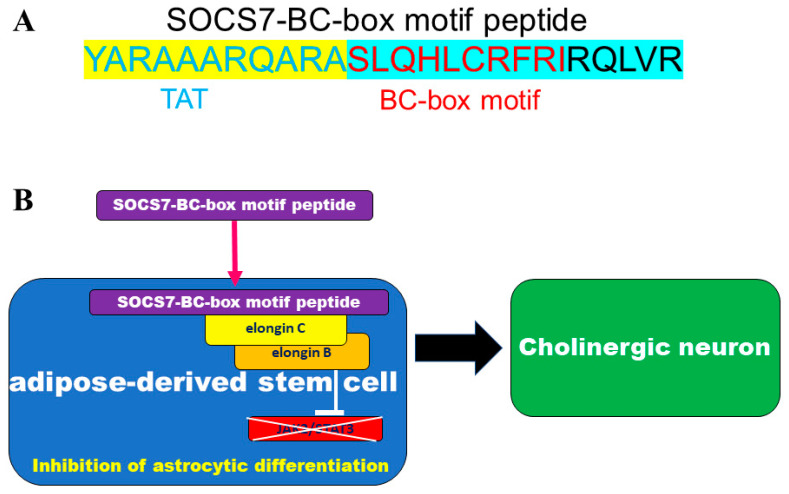
(**A**) Structure of protein transduction domain (TAT)-conjugated SOCS7-derived-BC box motif peptide. (**B**) Molecular mechanism of SOCS7-derived BC-box motif peptide-mediated neuronal differentiation of adipose-derived stem cells. It is suggested that cholinergic differentiation of ADMSC occurs after the SOCS7-BC box motif peptide bonds to elongin C and JAK2 is ubiquitinated.

## Data Availability

The data presented in this study are openly available only in this text.

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
