# Peer review of "SOCS7-Derived BC-Box Motif Peptide Mediated Cholinergic Differentiation of Human Adipose-Derived Mesenchymal Stem Cells"

_ijms, 2023, doi:10.3390/ijms24032786_

Round 1
Reviewer 1 Report
To the authors,
This study demonstrates with straightforward design the differentiation of adipose-derived mesenchymal stem cells (ADMSC) to Cholinergic Phenotype neurons using specific peptide SOCS7.
There are some crucial issues that the authors must analyze before acceptance.
Nevertheless, some points need to discuss.
Some genes and proteins were lacking in the Western Blot analysis, like nestin, Beta-tubulin III and to demonstrate the pluripotency of the cells by trilineage differentiation ADMSC as well that immature markers that could be presented in mesenchymal cells and not presented in differentiated conditions. The authors could be better at discussing that.
Introduction
1. It lacks some approach to differentiation proteins that lack expression when the cells differentiate into neuronal phenotypes.
Methods
1. ADMSC and those immature markers that could be presented in mesenchymal cells and not in differentiated conditions. The authors could be better discussing that in the Discussion topic.
2. There is some lack of the proteins (Nestin, YAP, AMOT, Beta Tubulin III), and in the Western Blot analysis of OCT4, NANOG, FGF2, REX1, VEGF, ABCG2, TUBB3, MAP2, NEFM SOX2, NES, SLC1A3, S100, and ASCL1 genes, when the research is in differentiation development. The authors could be presenting that in the Introduction topic.
Results
1.The authors have identified the ADMSC with Cytometric Analysis. Nevertheless, there is not a demonstration of the pluripotency of the cells by trilineage differentiation. It was mandatory by the International Society for Cell Therapy (Cytotherapy 2006, 8, 315–317. https://doi.org/10.1080/14653240600855905, Cytotherapy 2013, 15(6), 641-8. https://doi.org/10.1016/j.jcyt.2013.02.006.
2. This study demonstrated Western Blot and immunocytochemistry methods based on the affirmation of the Cholinergic Phenotype. The authors do not demonstrate the functionality of the Cholinergic Phenotype neurons with Acetylcholine production. Explain that.
Discussion
The authors need to compare with authors that obtained the same differentiation with other protocols.
Author Response
To Reviewer 1
1) Some genes and proteins were lacking in the Western Blot analysis, like nestin, Beta-tubulin III and to demonstrate the pluripotency of the cells by trilineage differentiation ADMSC as well that immature markers that could be presented in mesenchymal cells and not presented in differentiated conditions. The authors could be better at discussing that.
→ Thank you for the comment. We additionally performed Western blot analysis for nestin, β-tubulin â…¢, and MAP2 (Figure 5). Furthermore, to demonstrate the pluripotency we preformed fluorescence-immunocytochemistry for naïve ADMSCs (Figure 3). At “introduction” and “discussion”, trilineage differentiation was additionally explained.
2) It lacks some approach to differentiation proteins that lack expression when the cells differentiate into neuronal phenotypes.→As above described, protein expression study was added.
3) ADMSC and those immature markers that could be presented in mesenchymal cells and not in differentiated conditions. The authors could be better discussing that in the Discussion topic.
→ Similarly, as above described, we added fluorescence-immunocytochemical study for naïve ADMSCs (Figure 3).
4) There is some lack of the proteins (Nestin, YAP, AMOT, Beta Tubulin III), and in the Western Blot analysis of OCT4, NANOG, FGF2, REX1, VEGF, ABCG2, TUBB3, MAP2, NEFM SOX2, NES, SLC1A3, S100, and ASCL1 genes, when the research is in differentiation development. The authors could be presenting that in the Introduction topic.
→ At “introduction”, description about expressions of proteins and genes of ADMSCs was added.
5) The authors have identified the ADMSC with Cytometric Analysis. Nevertheless, there is not a demonstration of the pluripotency of the cells by trilineage differentiation. It was mandatory by the International Society for Cell Therapy (Cytotherapy 2006, 8, 315–317. https://doi.org/10.1080/14653240600855905, Cytotherapy 2013, 15(6), 641-8. https://doi.org/10.1016/j.jcyt.2013.02.006.
→ “Cytotherapy 2006, 8, 315–317. “ and “Cytotherapy 2013, 15(6), 641-8. “ were added in “Refences”.
6) This study demonstrated Western Blot and immunocytochemistry methods based on the affirmation of the Cholinergic Phenotype. The authors do not demonstrate the functionality of the Cholinergic Phenotype neurons with Acetylcholine production. Explain that.
→ Since demonstration for direct acetylcholine production is difficult, we employed immunocytochemistry for acetylcholine transferase that is metabolizing enzyme of acetylcholine
7) The authors need to compare with authors that obtained the same differentiation with other protocols. 
→ The cholinergic differentiation with other protocols was added in “Discussion”.

Reviewer 2 Report
The authors developed a novel method of neuronal differentiation of ADMSCs into cholinergic phenotype using SOCS7 peptide, which may be important for neuronal regeneration in motor neuron diseases.
Comments
1. Line 19: Osteoblasts should be included to cell types, which differentiated from ADMSCs.
2. Figs 1, 4: Magnification bars should be included.
3. Fig 2 is missing. This should be corrected.
4. Line 134: All the typos should be corrected.
5. Section 2.3: The description of peptide choice and the choice of its concentration should be presented in detail.
6. Fig 5: A and B parts should be indicated.
Author Response
To Reviewer 2.
- Line 19: Osteoblasts should be included to cell types, which differentiated from ADMSCs.
→ Osteoblast was added.
- Figs 1, 4: Magnification bars should be included.
→ Scale bars were added.
- Fig 2 is missing. This should be corrected.
→ We corrected it. Number of figures amounts to 7.
- Line 134: All the types should be corrected.
→ All the types were corrected.
- Section 2.3: The description of peptide choice and the choice of its concentration should be presented in detail.
→ The description of peptide choice and the choice of its concentration were added.
- Fig 5: A and B parts should be indicated.
→ They were indicated.
